# Black-Box Adversarial Attack with Transferable Model-based Embedding

**Zhichao Huang, Tong Zhang**
The Hong Kong University of Science and Technology
`zhuangbx@connect.ust.hk, tongzhang@tongzhang-ml.org`

## Abstract

We present a new method for black-box adversarial attack. Unlike previous methods that combined transfer-based and scored-based methods by using the gradient or initialization of a surrogate white-box model, this new method tries to learn a low-dimensional embedding using a pretrained model, and then performs efficient search within the embedding space to attack an unknown target network. The method produces adversarial perturbations with high level semantic patterns that are easily transferable. We show that this approach can greatly improve the query efficiency of black-box adversarial attack across different target network architectures. We evaluate our approach on MNIST, ImageNet and Google Cloud Vision API, resulting in a significant reduction on the number of queries. We also attack adversarially defended networks on CIFAR10 and ImageNet, where our method not only reduces the number of queries, but also improves the attack success rate.

## 1 Introduction

The wide adoption of neural network models in modern applications has caused major security concerns, as such models are known to be vulnerable to adversarial examples that can fool neural networks to make wrong predictions (Szegedy et al., 2014). Methods to attack neural networks can be divided into two categories based on whether the parameters of the neural network are assumed to be known to the attacker: white-box attack and black-box attack. There are several approaches to find adversarial examples for black-box neural networks. The transfer-based attack methods first pretrain a source model and then generate adversarial examples using a standard white-box attack method on the source model to attack an unknown target network (Goodfellow et al., 2015; Madry et al., 2018; Carlini & Wagner, 2017; Papernot et al., 2016a). The score-based attack requires a loss-oracle, which enables the attacker to query the target network at multiple points to approximate its gradient. The attacker can then apply the white-box attack techniques with the approximated gradient (Chen et al., 2017; Ilyas et al., 2018a; Tu et al., 2018).

A major problem of the transfer-based attack is that it can not achieve very high success rate. And transfer-based attack is weak in targeted attack. On the contrary, the success rate of score-based attack has only small gap to the white-box attack but it requires many queries. Thus, it is natural to combine the two black-box attack approaches, so that we can take advantage of a pretrained white-box source neural network to perform more efficient search to attack an unknown target black-box model.

In fact, in the recent NeurIPS 2018 Adversarial Vision Challenge (Brendel et al., 2018), many teams transferred adversarial examples from a source network as the starting point to carry out black-box boundary attack (Brendel et al., 2017). $\mathcal{N}$Attack also used a regression network as initialization in the score-based attack (Li et al., 2019a). The transferred adversarial example could be a good starting point that lies close to the decision boundary for the target network and accelerate further optimization. P-RGF (Cheng et al., 2019) used the gradient information from the source model to accelerate searching process. However, gradient information is localized and sometimes it is misleading. In this paper, we push the idea of using a pretrained white-box source network to guide black-box attack significantly further, by proposing a method called TRansferable EMbedding based Black-box Attack (TREMBA). TREMBA contains two stages: (1) train an encoder-decoder that can effectively generate adversarial perturbations for the source network with a low-dimensional embedding space; (2) apply NES (Natural Evolution Strategy) of (Wierstra et al., 2014) to the

low-dimensional embedding space of the pretrained generator to search adversarial examples for the target network. TREMBA uses global information of the source model, capturing high level semantic adversarial features that are insensitive to different models. Unlike noise-like perturbations, such perturbations would have much higher transferablity across different models. Therefore we could gain query efficiency by performing queries in the embedding space.

We note that there have been a number of earlier works on using generators to produce adversarial perturbations in the white-box setting (Baluja & Fischer, 2018; Xiao et al., 2018; Wang & Yu, 2019). While black-box attacks were also considered there, they focused on training generators with dynamic distillation. These early approaches required many queries to fine-tune the classifier for different target networks, which may not be practical for real applications. While our approach also relies on a generator, we train it as an encoder-decoder that produces a low-dimensional embedding space. By applying a standard black-box attack method such as NES on the embedding space, adversarial perturbations can be found efficiently for a target model.

It is worth noting that the embedding approach has also been used in AutoZOOM (Tu et al., 2018). However, it only trained the autoencoder to reconstruct the input, and it did not take advantage of the information of a pretrained network. Although it also produces structural perturbations, these perturbations are usually not suitable for attacking regular networks and sometimes its performance is even worse than directly applying NES to the images (Cheng et al., 2019; Guo et al., 2019). TREMBA, on the other hand, tries to learn an embedding space that can efficiently generate adversarial perturbations for a pretrained source network. Compared to AutoZOOM, our new method produces adversarial perturbation with high level semantic features that could hugely affect arbitrary target networks, resulting in significantly lower number of queries.

We summarize our contributions as follows:

1. We propose TREMBA, an attack method that explores a novel way to utilize the information of a pretrained source network to improve the query efficiency of black-box attack on a target network.

2. We show that TREMBA can produce adversarial perturbations with high level semantic patterns, which are effective across different networks, resulting in much lower queries on MNIST and ImageNet especially for the targeted attack that has low transferablity.

3. We demonstrate that TREMBA can be applied to SOTA defended models (Madry et al., 2018; Xie et al., 2018). Compared with other black-box attacks, TREMBA increases success rate by approximately 10% while reduces the number of queries by more than 50%.

## 2 RELATED WORKS

There have been a vast literature on adversarial examples. We will cover the most relevant topics including white-box attack, black-box attack and defense methods.

**White-Box Attack** White-box attack requires the full knowledge of the target model. It was first discovered by (Szegedy et al., 2014) that adversarial examples could be found by solving an optimization problem with L-BFGS (Nocedal, 1980). Later on, other methods were proposed to find adversarial examples with improved success rate and efficiency (Goodfellow et al., 2015; Kurakin et al., 2016; Papernot et al., 2016b; Moosavi-Dezfooli et al., 2016). More recently, it was shown that generators can also construct adversarial noises with high success rate (Xiao et al., 2018; Baluja & Fischer, 2018).

**Black-Box Attack** Black-box attack can be divided into three categories: transfer-based, score-based and decision-based. It is well known that adversaries have high transferablity across different networks (Papernot et al., 2016a). Transfer-based methods generate adversarial noises on a source model and then transfer it to an unknown target network. It is known that targeted attack is harder than untargeted attack for transfer-based methods, and using an ensemble of source models can improve the success rate (Liu et al., 2016). Score-based attack assumes that the attacker can query the output scores of the target network. The attacker usually uses sampling methods to approximate the true gradient (Chen et al., 2017; Ilyas et al., 2018a; Li et al., 2019a; Chen et al., 2018). AutoZOOM tried to improve the query efficiency by reducing the sampling space with a bilinear transformation or an autoencoder (Tu et al., 2018). (Ilyas et al., 2018b) incorporated data and time prior to accelerate attacking. In

contrast to the gradient based method, (Moon et al., 2019) used combinatorial optimization to achieve good efficiency. In decision-based attack, the attacker only knows the output label of the classifier. Boundary attack and its variants are very powerful in this setting (Brendel et al., 2017; Dong et al., 2019). In NeutIPS 2018 Adversarial Vision Challenge (Brendel et al., 2018), some teams combined transfer-based attack and decision-based attack in their attacking methods (Brunner et al., 2018). And in a similar spirit, $\mathcal{N}$Attack also used a regression network as initialization in score-based attack (Li et al., 2019a). Gradient information from the surrogate model could also be used to accelerate the scored-based attack (Cheng et al., 2019) .

**Defense Methods** Several methods have been proposed to overcome the vulnerability of neural networks. Gradient masking based methods add non-differential operations in the model, interrupting the backward pass of gradients. However, they are vulnerable to adversarial attacks with the approximated gradient (Athalye et al., 2018; Li et al., 2019a). Adversarial training is the SOTA method that can be used to improve the robustness of neural networks. Adversarial training is a minimax game. The outside minimizer performs regular training of the neural network, and the inner maximizer finds a perturbation of the input to attack the network. The inner maximization process can be approximated with FGSM (Goodfellow et al., 2015), PGD (Madry et al., 2018), adversarial generator (Wang & Yu, 2019) etc. Moreover, feature denoising can improve the robustness of neural networks on ImageNet (Xie et al., 2018).

## 3 BLACK-BOX ADVERSARIAL ATTACK WITH GENERATOR

Consider a DNN classifier $F(x)$. Let $x \in [0, 1]^{\dim(x)}$ be an input, and let $F(x)$ be the output vector obtained before the softmax layer. We denote $F(x)_i$ as the $i$-th component for the output vector and $y$ as the label for the input. For un-targeted attack, our goal is to find a small perturbation $\delta$ such that the classifier predicts the wrong label, i.e. $\arg\max F(x + \delta) \neq y$. And for targeted attack, we want the classifier to predicts the target label $t$, i.e. $\arg\max F(x + \delta) = t$. The perturbation $\delta$ is usually bounded by $\ell_p$ norm: $\|\delta\|_p \leq \varepsilon$, with a small $\varepsilon > 0$.

Adversarial perturbations often have high transferablity across different DNNs. Given a white-box source DNN $F_s$ with known architecture and parameters, we can transfer its white-box adversarial perturbation $\delta_s$ to a black-box target DNN $F_t$ with reasonably good success rate. It is known that even if $x + \delta_s$ fails to be an adversarial example, $\delta_s$ can still act as a good starting point for searching adversarial examples using a score-based attack method. This paper shows that the information of $F_s$ can be further utilized to train a generator, and performing search on its embedding space leads to more efficient black-box attacks of an unknown target network $F_t$.

### 3.1 GENERATING ADVERSARIAL PERTURBATIONS WITH GENERATOR

Adversarial perturbations can be generated by a generator network $\mathcal{G}$. We explicitly divide the generator into two parts: an encoder $\mathcal{E}$ and a decoder $\mathcal{D}$. The encoder takes the origin input $x$ and output a latent vector $z = \mathcal{E}(x)$, where $\dim(z) \ll \dim(x)$. The decoder takes $z$ as the input and outputs an adversarial perturbation $\delta = \varepsilon \tanh(\mathcal{D}(z))$ with $\dim(\delta) = \dim(x)$. In our new method, we will train the generator $\mathcal{G}$ so that $\delta = \varepsilon \tanh(\mathcal{G}(x))$ can fool the source network $F_s$.

Suppose we have a training set $\{(x_1, y_1), \ldots, (x_n, y_n)\}$, where $x_i$ denotes the input and $y_i$ denotes its label. For un-targeted attack, we train the desired generator by minimizing the hinge loss used in the C&W attack (Carlini & Wagner, 2017):

$$\mathcal{L}_{\text{untarget}}(x_i, y_i) = \max\left( F_s(\varepsilon \tanh(\mathcal{G}(x_i)) + x_i)_{y_i} - \max_{j \neq y_i} F_s(\varepsilon \tanh(\mathcal{G}(x_i)) + x_i)_j, -\kappa \right), \quad (1)$$

And for targeted, we use

$$\mathcal{L}_{\text{target}}(x_i, t) = \max\left( \max_{j \neq t} F_s(\varepsilon \tanh(\mathcal{G}(x_i)) + x_i)_j - F_s(\varepsilon \tanh(\mathcal{G}(x_i)) + x_i)_t, -\kappa \right), \quad (2)$$

where $t$ denotes the targeted class and $\kappa$ is the margin parameter that can be used to adjust transferability of the generator. A higher value of $\kappa$ leads to higher transferability to other models (Carlini & Wagner, 2017). We focus on $\ell_\infty$ norm in this work. By adding point-wise tanh function to an unnormalized output $\mathcal{D}(z)$, and scaling it with $\varepsilon$, $\delta = \varepsilon \tanh(\mathcal{D}(z))$ is already bounded as $\|\delta\|_\infty < \varepsilon$.

Therefore we employ this transformation, so that we do not need to impose the infinity norm constraint explicitly. While hinge loss is employed in this paper, we believe other loss functions such the cross entropy loss will also work.

## 3.2 SEARCH OVER LATENT SPACE WITH NES

Given a new black-box DNN classifier $F_t(x)$, for which we can only query its output at any given point $x$. As in (Ilyas et al., 2018a; Wierstra et al., 2014), we can employ NES to approximate the gradient of a properly defined surrogate loss in order to find an adversarial example. Denote the surrogate loss by $\mathcal{L}$, rather than calculating $\nabla_\delta \mathcal{L}(x+\delta, y)$ directly, NES update $\delta$ by using $\nabla_\delta \mathbb{E}_{\omega \sim \mathcal{N}(\delta, \sigma^2)}[L(x+\omega, y)]$, which can be transformed into $\mathbb{E}_{\omega \sim \mathcal{N}(\delta, \sigma^2)}[L(x+\omega, y)\nabla_\omega \log(\mathcal{N}(\omega|\delta, \sigma^2))]$. The expectation can be approximated by taking finite samples. And we could use the following equation to iteratively update $\delta$:

$$\delta_{t+1} = \prod_{[-\varepsilon, \varepsilon]} (\delta_t - \eta \cdot \text{sign}(\frac{1}{b}\sum_{k=1}^{b} \mathcal{L}(x+\omega_k, y)\nabla \log \mathcal{N}(\omega_k|\delta_t, \sigma^2))), \quad (3)$$

where $\eta$ is the learning rate, $b$ is the minibatch sample size, $\omega_k$ is the sample from the gaussian distribution and $\prod_{[-\varepsilon, \varepsilon]}$ represents a clipping operation, which projects $\delta$ onto the $\ell_\infty$ ball. The sign function provides an approximation of the gradient, which has been widely used in adversarial attack (Ilyas et al., 2018a; Madry et al., 2018). However, it is observed that more effective attacks can be obtained by removing the sign function (Li et al., 2019b). Therefore in this work, we remove the sign function from Eqn (3) and directly use the estimated gradient.

Instead of performing search on the input space, TREMBA performs search on the embedding space $z$. The generator $\mathcal{G}$ explores the weakness of the source DNN $F_s$ so that $\mathcal{D}$ produces perturbations that can effective attack $F_s$. For a different unknown target network $F_t$, we show that our method can still generate perturbations leading to more effective attack of $F_t$. Given an input $x$ and its label $y$, we choose a starting point $z^0 = \mathcal{E}(x)$. The gradient of $z^t$ given by NES can be estimated as:

$$\nabla_{z^t} \mathcal{L}(x + \varepsilon \tanh(\mathcal{D}(z^t)), y) \approx \nabla_{z^t} \mathbb{E}_{\nu \sim \mathcal{N}(z^t, \sigma^2)} [\mathcal{L}(x + \varepsilon \tanh(\mathcal{D}(\nu)), y)] \quad (4)$$

$$\approx \frac{1}{b}\sum_{k=1}^{b} \mathcal{L}(x + \varepsilon \tanh(\mathcal{D}(\nu_k)), y)\nabla_{z^t} \log \mathcal{N}(\nu_k|z^t, \sigma^2).$$

where $\nu_k$ is the sample from the gaussian distribution $\mathcal{N}(z^t, \sigma^2)$. Moreover, $z^t$ is updated with stochastic gradient descent. The detailed procedure is presented in Algorithm 1. We do not need to do projection explicitly since $\delta$ already satisfies $\|\delta\|_\infty < \varepsilon$.

Next we shall briefly explain why applying NES on the embedding space $z$ can accelerate the search process. Adversarial examples can be viewed as a distribution lying around a given input. Usually this distribution is concentrated on some small regions, making the search process relatively slow. After training on the source network, the adversarial perturbations of TREMBA would have high level semantic patterns that are likely to be adversarial patterns of the target network. Therefore searching over $z$ is like searching adversarial examples in a lower dimensional space containing likely adversarial patterns. The distribution of adversarial perturbations in this space is much less concentrated. It is thus much easier to find effective adversarial patterns in the embedding space.

## 4 EXPERIMENTS

We evaluated the number of queries versus success rate of TREMBA on undefended network in two datasets: MNIST (LeCun et al., 1998) and ImageNet (Russakovsky et al., 2015). Moreover, we evaluated the efficiency of our method on adversarially defended networks in CIFAR10 (Krizhevsky & Hinton, 2009) and ImageNet. We also attacked Google Cloud Vision API to show TREMBA can generalize to truly black-box model.[1] We used the hinge loss from Eqn 1 and 2 as the surrogate loss for un-targeted and targeted attack respectively.

We compared TREMBA to four methods: (1) **NES**: Method introduced by (Ilyas et al., 2018a), but without the sign function for reasons explained earlier. (2) **Trans-NES**: Take an adversarial

---

[1]Our code is available at `https://github.com/TransEmbedBA/TREMBA`

---

**Algorithm 1** Black-Box adversarial attack on the embedding space

---

**Input:**

Target Network $F_t$; Input $x$ and its label $y$ or the target class $t$; Encoder $\mathcal{E}$; Decoder $\mathcal{D}$; Standard deviation $\sigma$; Learning rate $\eta$; Sample size $b$; Iterations $T$; Bound for adversarial perturbation $\varepsilon$

**Output:** Adversarial perturbation $\delta$

1: $z_0 = \mathcal{E}(x)$
2: **for** $t = 1$ to $T$ **do**
3:     Sample Gaussian noise $\nu_1, \nu_2, \cdots, \nu_b \sim \mathcal{N}(z_{t-1}, \sigma^2)$
4:     Calculate $\mathcal{L}_i = \mathcal{L}_{\text{untarget}}(x, y)$ or $\mathcal{L}_{\text{target}}(x, t)$
5:     Update $z_t = z_{t-1} - \frac{\eta}{b} \sum_{i=1}^{b} \mathcal{L}_i \nabla_{z_{t-1}} \log \mathcal{N}(\nu_i | z_{t-1}, \sigma^2)$
6: **end for**
7: **return** $\delta = \varepsilon \tanh(\mathcal{D}(z_T))$

---

perturbation generated by PGD or FGSM on the source model to initialize NES. (3) **AutoZOOM**: Attack target network with an unsupervised autoencoder described in (Tu et al., 2018). For fair comparisons with other methods, the strategy of choosing sample size was removed. (4) **P-RGF**: Prior-guided random gradient-free method proposed in (Cheng et al., 2019). The P-RGF$_{\text{D}}(\lambda^*)$ version was compared. We also combined P-RGF with initialization from Trans-NES$_{\text{PGD}}$ to form a more efficient method for comparison, denoted by Trans-P-RGF.

Since different methods achieve different success rates, we need to compare their efficiency at different levels of success rate. For method $i$ with success rate $s_i$, the average number of queries is $q_i$ for all success examples. Let $q^*$ denote the upper limit of queries, we modified the average number of queries to be $q_i^* = [(\max_j s_j - s_i) \cdot q^* + s_i \cdot q_i] / \max_j s_j$, which unified the level of success rate and treated queries of failure examples as the upper limit on the number of queries. Average queries sometimes could be misleading due to the the heavy tail distribution of queries. Therefore we plot the curve of success rate at different query levels to show the detailed behavior of different attacks.

The upper limit on the number of queries was set to 50000 for all datasets, which already gave very high success rate for nearly all the methods. Only correctly classified images were counted towards success rate and average queries. And to fairly compare these methods, we chose the sample size to be the same for all methods. We also added momentum and learning decay for optimization. And we counted the queries as one if its starting point successfully attacks the target classifier. The learning rate was fine-tuned for all algorithms. We listed the hyperparameters and architectures of generators and classifiers in Appendix B and C.

## 4.1 BLACK-BOX ATTACK ON MNIST

We trained four neural networks on MNIST, denoted by ConvNet1, ConvNet1*, ConvNet2 and FCNet. ConvNet1* and ConvNet1 have the same architecture but different parameters. All the network achieved about 99% accuracy. The generator $\mathcal{G}$ was trained on ConvNet1* using all images from the training set. Each attack was tested on images from the MNIST test set. The limit of $\ell_\infty$ was $\varepsilon = 0.2$.

We performed un-targeted attack on MNIST. Table 1 lists the success rate and the average queries. Although the success rate of TREMBA is slightly lower than Trans-NES in ConvNet1 and FCNet, their success rate are already close to 100% and TREMBA achieves about 50% reduction of queries compared with other attacks. In contrast to efficient attack on ImageNet, P-RGF and Trans-P-RGF behaves very bad on MNIST. Figure 4.1 shows that TREMBA consistently achieves higher success rate at nearly all query levels.

## 4.2 BLACK-BOX ATTACK ON IMAGENET

We randomly divided the ImageNet validation set into two parts, containing 49000 and 1000 images respectively. The first part was used as the training data for the generator $\mathcal{G}$, and the second part was used for evaluating the attacks. We evaluated the efficiency of all adversarial attacks on VGG19 (Simonyan & Zisserman, 2014), Resnet34 (He et al., 2016), DenseNet121 (Huang et al., 2017) and MobilenetV2 (Sandler et al., 2018). All networks were downloaded using *torchvision* package. We set $\varepsilon = 0.03125$.

Table 1: Success rate and average queries of un-targeted attack on MNIST. $\varepsilon = 0.2$

| Attack | ConvNet1 | | ConvNet2 | | FCNet | |
|---|---|---|---|---|---|---|
| | Success | Queries | Success | Queries | Success | Queries |
| NES | 97.88% | 4380 | 90.32% | 5428 | 99.98% | 1183 |
| Trans-NES$_{PGD}$ | **98.65%** | 2113 | 90.22% | 4691 | **99.99%** | 818 |
| Trans-NES$_{FGSM}$ | 98.34% | 3592 | 91.32% | 4218 | 99.99% | 1540 |
| AutoZOOM | 93.39% | 5874 | 91.21% | 2645 | 99.69% | 823 |
| P-RGF | 68.53% | 16135 | 39.85% | 29692 | 90.42% | 8289 |
| Trans-P-RGF | 66.34% | 16428 | 27.57% | 35576 | 68.39% | 18818 |
| TREMBA | 98.00% | **1064** | **92.63%** | **1359** | 99.75% | **470** |

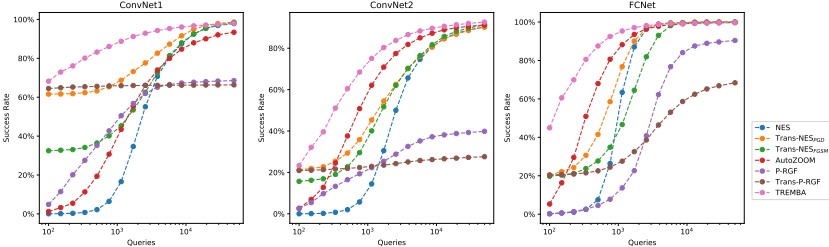

Figure 1: Success rate of un-targeted attack at different query levels for undefended MNIST models.

Following (Liu et al., 2016), we used an ensemble (VGG16, Resnet18, Squeezenet (Iandola et al., 2016) and Googlenet (Szegedy et al., 2015)) as the source model to improve transferablity (Liu et al., 2016) for both targeted and un-targeted attack. TREMBA, Trans-NES, P-RGF and Trans-P-RGF all used the same source model for fair comparison. We chose several target class. Here, we show the result of attacking class 0 (tench) in Table 2 and Figure 2. And we leave the result of attacking other classes in Appendix A.1. The average queries for TREMBA is about 1000 while nearly all the average queries for other methods are more than 6000. TREMBA also achieves much lower queries for un-targeted attack on ImageNet. The result is shown in Appendix A.2 due to space limitation. And we also compared TREMBA with CombOpt (Moon et al., 2019) in the Appendix A.9.

Figure 3 shows the adversarial perturbations of different methods. Unlike adversarial perturbations produced by PGD, the perturbations of TREMBA reflect some high level semantic patterns of the targeted class such as the fish scale. As neural networks usually capture such patterns for classification, the adversarial perturbation of TREMBA would be more easy to transfer than the noise-like perturbation produced by PGD. Therefore TREMBA can search very effectively for the target network. More examples of perturbations of TREMBA are shown in Appendix A.3.

**Choice of ensemble:** We performed attack on different ensembles of source model, which is shown in Appendix A.4. TREMBA outperforms the other methods in different ensemble model. And more source networks lead to better transferability for TREMBA, Trans-NES and Trans-P-RGF.

**Varying $\varepsilon$:** We also changed $\varepsilon$ and performed attack on $\varepsilon = 0.02$ and $\varepsilon = 0.04$. As shown in Appendix A.5, TREMBA still outperforms the other methods despite using the $\mathcal{G}$ trained on $\varepsilon = 0.03125$. We also show the result of TREMBA for commonly used $\varepsilon = 0.05$.

**Sample size and dimension the embedding space:** To justify the choice of sample size, we performed a hyperparameter sweep over $b$ and the result is shown in Appendix A.6. And we also changed the dimension of the embedding space for AutoZOOM and Trans-P-RGF. As shown in Appendix A.7, the performance gain of TREMBA does not purely come from the diminishing of dimension of the embedding space.

## 4.3 Black-Box Attack on Defended Models

This section presents the results for attacking defended networks. We performed un-targeted attack on two SOTA defense methods on CIFAR10 and ImageNet. MNIST is not studied since it is already

Table 2: Success rate and average queries of black-box targeted attack on ImageNet. Targeted class is class 0 (tench). $\varepsilon = 0.03125$

| Attack | VGG19 | | Resnet34 | | DenseNet121 | | MobilenetV2 | |
|---|---|---|---|---|---|---|---|---|
| | Success | Queries | Success | Queries | Success | Queries | Success | Queries |
| NES | 94.86% | 12283 | 93.89% | 14418 | 95.65% | 12538 | 97.76% | 10276 |
| Trans-NES$_{PGD}$ | 96.26% | 6854 | 95.97% | 8737 | 96.59% | 8627 | 98.04% | 9375 |
| Trans-NES$_{FGSM}$ | 90.85% | 12885 | 91.81% | 14090 | 93.61% | 12859 | 97.48% | 9983 |
| AutoZOOM | 25.80% | 40195 | 26.25% | 39681 | 31.98% | 37628 | 27.03% | 39689 |
| P-RGF | 96.12% | 6951 | 90.28% | 10221 | 91.84% | 11563 | 88.94% | 14596 |
| Trans-P-RGF | 98.06% | 2262 | 93.61% | 6309 | 94.69% | 7263 | 91.60% | 10048 |
| TREMBA | **98.47%** | **853** | **96.38%** | **1206** | **98.50%** | **1124** | **99.16%** | **1210** |

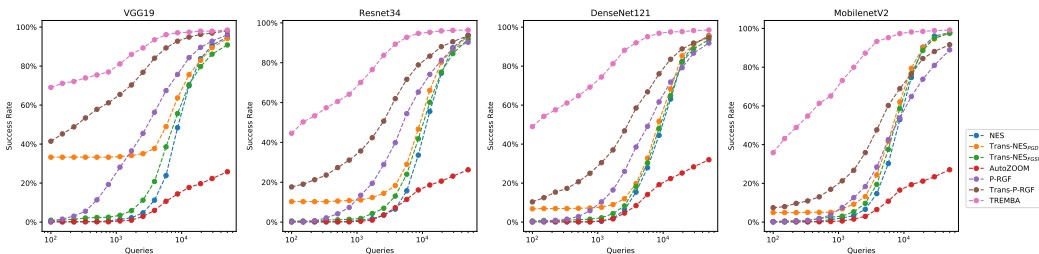

Figure 2: The success rate of black-box adversarial targeted attack at different query levels for ImageNet models. The targeted class is tench

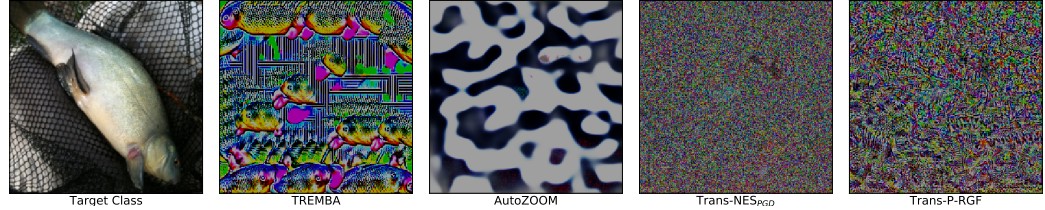

Figure 3: Visualization of adversarial perturbations targeted at tench

robust against very strong white-box attacks. For CIFAR10, the defense model was going through PGD minimax training (Madry et al., 2018). We directly used their model as the source network[2], denoted by WResnet. To test whether these methods can transfer to a defended network with a different architecture, we trained a defended ResNeXt (Xie et al., 2017) using the same method. For ImageNet, we used the SOTA model[3] from (Xie et al., 2018). We used "ResNet152 Denoise" as the source model and transfered adversarial perturbations to the most robust "ResNeXt101 DenoiseAll". Following the previous settings, we set $\varepsilon = 0.03125$ for both CIFAR10 and ImageNet.

As shown in Table 3, TREMBA achieves higher success rates with lower number of queries. TREMBA achieves about $10\%$ improvement of success rate while the average queries are reduced by more than $50\%$ on ImageNet and by $80\%$ on CIFAR10. The curves in Figure 4(a) and 4(b) show detailed behaviors. The performance of AutoZOOM surpasses Trans-NES on defended models. We suspect that low-frequency adversarial perturbations produced by AutoZOOM will be more suitable to fool the defended models than the regular networks. However, the patterns learned by AutoZOOM are still worse than adversarial patterns learned by TREMBA from the source network.

---

[2] https://github.com/MadryLab/cifar10_challenge
[3] https://github.com/facebookresearch/ImageNet-Adversarial-Training

Table 3: Success rate of average queries of black-box un-targeted attack on defended CIFAR10 and ImageNet model. Source network is WResNet and ResNet152 Denoise.

| Attack | CIFAR10 ResneXt | | ImageNet RexneXt101 DenoiseAll | |
|---|---|---|---|---|
| | Success | Queries | Success | Queries |
| NES | 32.17% | 24521 | 29.72% | 26526 |
| Trans-NES$_{PGD}$ | 32.92% | 20735 | 32.84% | 20446 |
| Trans-NES$_{FGSM}$ | 33.17% | 20873 | 33.66% | 18547 |
| AutoZOOM | 33.70% | 14870 | 38.75% | 14605 |
| P-RGF | 22.37% | 25818 | 32.51% | 17926 |
| Trans-P-RGF | 20.88% | 27222 | 31.03% | 19262 |
| TREMBA | **42.73%** | **2528** | 49.59% | 5985 |
| TREMBA$_{OSP}$ | 41.56% | 4994 | **50.41%** | **4771** |

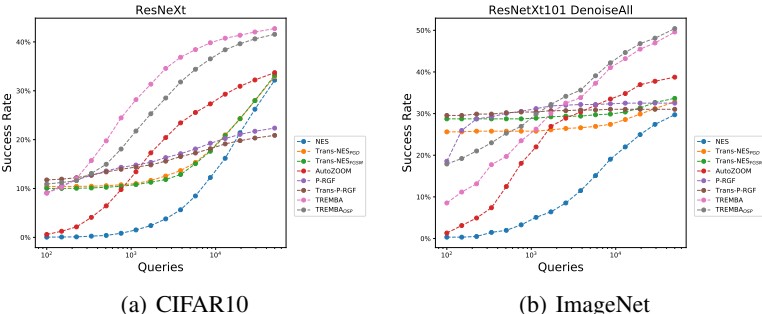

(a) CIFAR10          (b) ImageNet

Figure 4: The success rate at different query levels for defended CIFAR10 and ImageNet models. (a)CIFAR10; (b)ImageNet.

Table 4: Success rate and average queries of un-targeted attack of 10 images on Google Vision API.

| Method | NES | AutoZOOM | Trans-NES$_{PGD}$ | P-RGF | Trans-P-RGF | TREMBA |
|---|---|---|---|---|---|---|
| Success | 70.00% | 20.00% | 70.00% | 50.00% | 60.00% | 90.00% |
| Queries | 245 | 410 | 114 | 324 | 167 | 8 |

**An optimized starting point for TREMBA**: $z_0 = \mathcal{E}(x)$ is already a good starting point for attacking undefended networks. However, the capability of generator is limited for defended networks (Wang & Yu, 2019). Therefore, $z_0$ may not be the best starting point we can get from the defended source network. To enhance the usefulness of the starting point, we optimized $z$ on the source network by gradient descent and found

$$z_0^* = \arg\min_z \max\left( F_s(\varepsilon \tanh(\mathcal{D}(z)) + x)_y - \max_{j \neq y_i} F_s(\varepsilon \tanh(\mathcal{D}(z)) + x)_j, -\kappa \right). \quad (5)$$

The method is denoted by TREMBA$_{OSP}$ (TREMBA with optimized starting point). Figure 4 shows TREMBA$_{OSP}$ has higher success rate at small query levels, which means its starting point is better than TREMBA.

## 4.4 ATTACK GOOGLE CLOUD VISION API

We also attacked the Google Cloud Vision API, which was much harder to attack than the single neural network. Therefore we set $\varepsilon = 0.05$ and perform un-targeted attack on the API, changing the top1 label to whatever is not on top1 before. We chose 10 images for the ImageNet dataset and set query limit to be 500 due to high cost to use the API. As shown Table 4, TREMBA achieves much higher accuracy success rate and lower number of queries. We show the example of successfully attacked image in Appendix A.8.

## 5 CONCLUSION

We propose a novel method, TREMBA, to generate likely adversarial patterns for an unknown network. The method contains two stages: (1) training an encoder-decoder to generate adversarial perturbations for the source network; (2) search adversarial perturbations on the low-dimensional embedding space of the generator for any unknown target network. Compared with SOTA methods, TREMBA learns an embedding space that is more transferable across different network architectures. It achieves two to six times improvements in black-box adversarial attacks on MNIST and ImageNet and it is especially efficient in performing targeted attack. Furthermore, TREMBA demonstrates great capability in attacking defended networks, resulting in a nearly $10\%$ improvement on the attack success rate, with two to six times of reductions in the number of queries. TREMBA opens up new ways to combine transfer-based and score-based attack methods to achieve higher efficiency in searching adversarial examples.

For targeted attack, TREMBA requires different generators to attack different classes. We believe methods from conditional image generation (Mirza & Osindero, 2014) may be combined with TREMBA to form a single generator that could attack multiple targeted classes. We leave it as a future work.

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

## A    EXPERIMENT RESULT

### A.1    TARGETED ATTACK ON IMAGENET

Figure 9 shows result of the targeted attack on dipper, American chameleon, night snake, ruffed grouse and black swan. TREMBA achieves much higher success rate than other methods at almost all queries level.

### A.2    UN-TARGETED ATTACK ON IMAGENET

We used the same source model from targeted attack as the source model for un-targeted attack. We report our evaluation results in Table 5 and Figure 5. Compared with Trans-P-RGF, TREMBA reduces the number of queries by more than a half in ResNet34, DenseNet121 and MobilenetV2. Searching in the embedding space of generator remains very effective even when the target network architecture differs significantly from the networks in the source model.

Table 5: Success rate and average queries of un-targeted attack on ImageNet. $\varepsilon = 0.03125$

| Attack | VGG19 | | Resnet34 | | DenseNet121 | | MobilenetV2 | |
|---|---|---|---|---|---|---|---|---|
| | Success | Queries | Success | Queries | Success | Queries | Success | Queries |
| NES | **100%** | 924 | **100%** | 1255 | **100%** | 1235 | 99.86% | 872 |
| Trans-NES$_{PGD}$ | **100%** | 441 | **100%** | 827 | **100%** | 838 | **100%** | 733 |
| Trans-NES$_{FGSM}$ | **100%** | 586 | **100%** | 982 | **100%** | 961 | **100%** | 648 |
| AutoZOOM | 94.18% | 5184 | 96.25% | 3754 | 94.56% | 4567 | 95.38% | 4213 |
| P-RGF | **100%** | 277 | 99.72% | 635 | **100%** | 709 | 99.72% | 730 |
| Trans-P-RGF | **100%** | 130 | 99.86% | 371 | 99.18% | 806 | 99.86% | 522 |
| TREMBA | **100%** | **88** | **100%** | **183** | **100%** | **172** | **100%** | **61** |

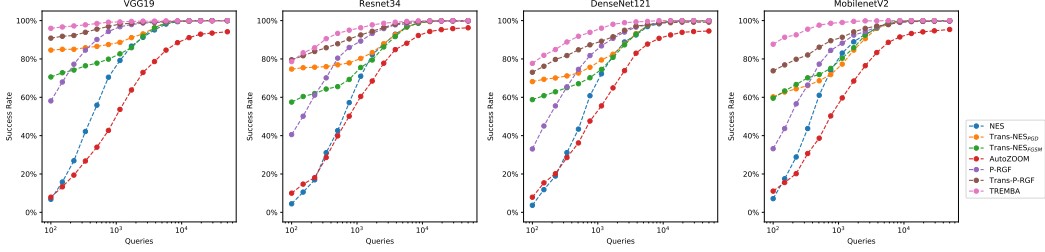

Figure 5: The success rate of un-targeted black-box adversarial attack at different query levels for undefended ImageNet models.

### A.3    VISUALIZATION OF TARGETED PERTURBATION

Figure 10 shows some examples of adversarial perturbations produced by TREMBA. The first column is one image of the target class and other columns are examples of perturbations (amplified by 10 times). It is easy to discover some features of the target class in the adversarial perturbation such as the feather for birds and the body for snakes.

### A.4    EXPERIMENTS ON DIFFERENT ENSEMBLES

We chose two more source ensemble models for evaluation. The first ensemble contains VGG16 and Squeezenet. And the second ensemble is consist of VGG16, Squeezenet and Googlenet. Figure 6 shows our result for targeted attack for ImageNet. We only compared Trans-NES$_{PGD}$ and Trans-P-RGF since they are the best variants from Trans-NES and P-RGF.

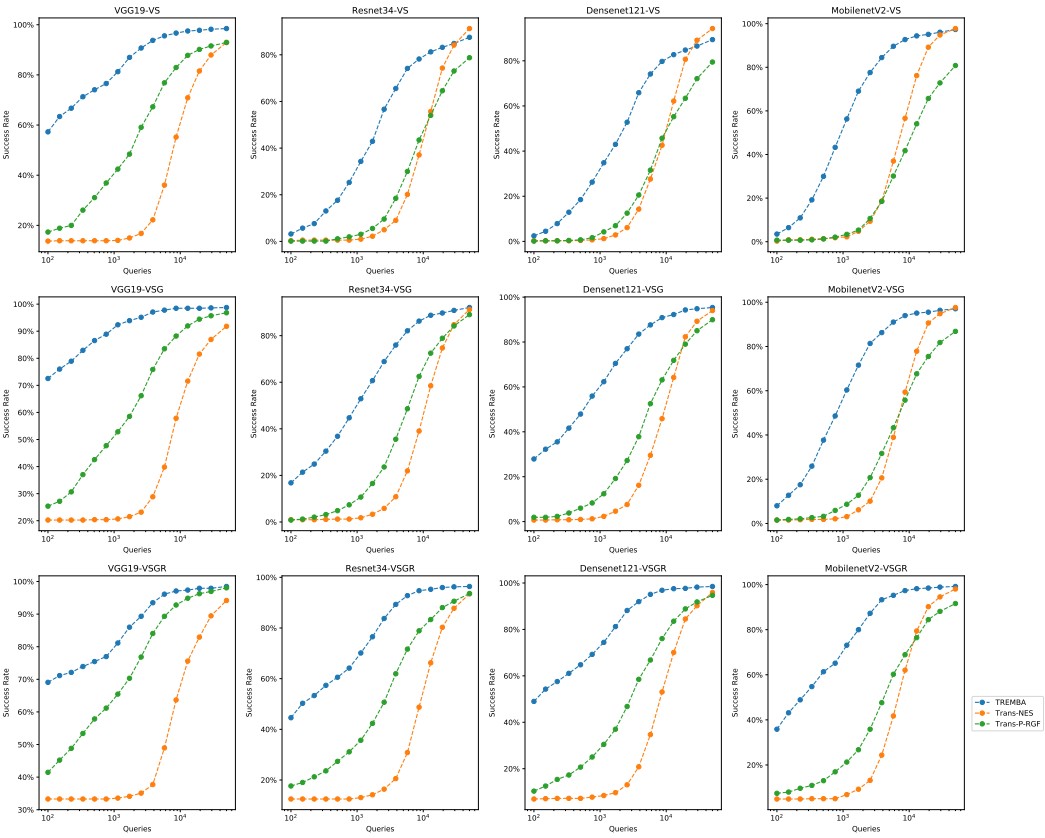

Figure 6: We show the success rate at different query levels for targeted attack for different ensemble source networks. V represents VGG16; S represents Squeezenet; G represents Googlenet; R represents Resnet18

### A.5 VARYING $\varepsilon$

We chose $\varepsilon = 0.02$ and $\varepsilon = 0.04$ and performed targeted attack on ImageNet. Although TREMBA used the same model that is trained on $\varepsilon = 0.03125$, it still outperformed other methods, which shows that TREMBA can also generalize to different strength of adversarial attack with different $\varepsilon$.

For the commonly used $\varepsilon = 0.05$, TREMBA also performs well. The results are shown in Table 6, Table 7, and Figure 8.

### A.6 VARYING SAMPLE SIZE

We performed a hyperparameter sweep over $b$ on Densenet121 on un-targeted attack on ImageNet. $b = 20$ may not be the best choice Trans-NES, but it is not the best for TREMBA, either. Generally, the performance is not very sensitive to $b$, and TREMBA will also outperform other methods even if we fine-tune the sample size for all the methods.

### A.7 DIMENSION OF THE EMBEDDING SPACE

We slightly changed the architecture of the autoencoder by adding max pooling layers and changing the number of filters and perform un-targeted attack on ImageNet. More specifically, we added additional max pooling layers after the first and the fourth convolution layers and changed the number of filters of the last layer in the encoder to be 8. Thus, the dimension of the embedding space would be $8 \times 8 \times 8$. And we also changed the factor of bilinear sampling in the decoder. The remaining

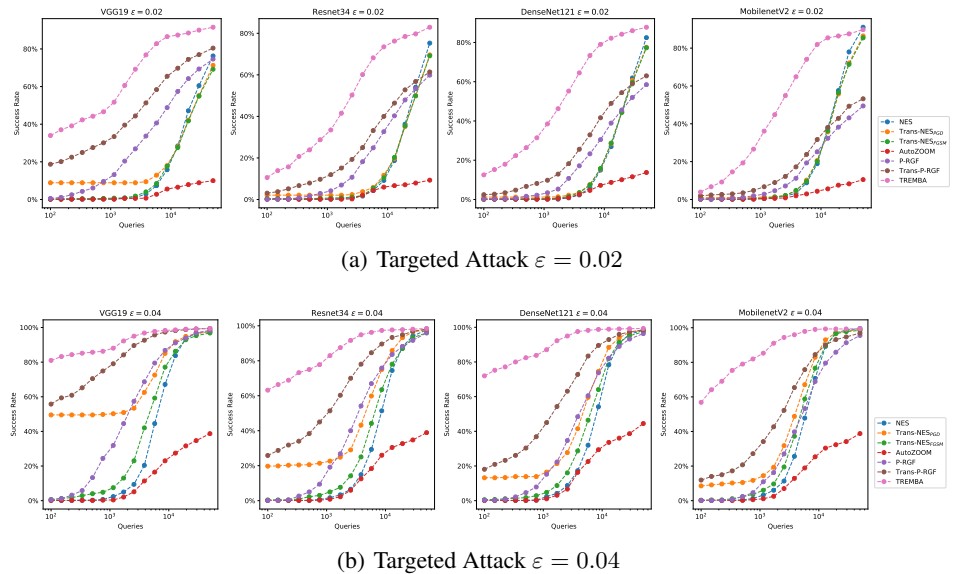

(a) Targeted Attack $\varepsilon = 0.02$

(b) Targeted Attack $\varepsilon = 0.04$

Figure 7: We show the success rate at different query levels for attack at different $\varepsilon$ for ImageNet.

Table 6: Success rate and average queries of un-targeted attack on ImageNet. $\varepsilon = 0.05$

| Attack | VGG19 | | Resnet34 | | DenseNet121 | | MobilenetV2 | |
|---|---|---|---|---|---|---|---|---|
| | Success | Queries | Success | Queries | Success | Queries | Success | Queries |
| NES | **100%** | 651 | **100%** | 850 | **100%** | 840 | 99.86% | 640 |
| Trans-NES$_{PGD}$ | **100%** | 74 | **100%** | 196 | **100%** | 235 | **100%** | 169 |
| Trans-NES$_{FGSM}$ | **100%** | 232 | **100%** | 401 | **100%** | 361 | **100%** | 272 |
| AutoZOOM | 99.72% | 1743 | 99.58% | 1481 | 99.32% | 1730 | 99.29% | 1672 |
| P-RGF | **100%** | 178 | **100%** | 328 | **100%** | 436 | **100%** | 402 |
| Trans-P-RGF | **100%** | 44 | 99.44% | 418 | 98.09% | 1049 | **100%** | 157 |
| TREMBA | **100%** | **8** | **100%** | **27** | **100%** | **19** | **100%** | **8** |

Table 7: Success rate and average queries of targeted attack on ImageNet. $\varepsilon = 0.05$

| Attack | VGG19 | | Resnet34 | | DenseNet121 | | MobilenetV2 | |
|---|---|---|---|---|---|---|---|---|
| | Success | Queries | Success | Queries | Success | Queries | Success | Queries |
| NES | 99.03% | 6364 | 98.89% | 8003 | 99.32% | 7525 | 99.72$ | 5610 |
| Trans-NES$_{PGD}$ | 99.31% | 1968 | 99.31% | 3549 | 99.46% | 3731 | 99.86% | 3223 |
| Trans-NES$_{FGSM}$ | 99.03% | 4997 | 98.33% | 7298 | 98.23% | 6874 | 99.16% | 5034 |
| AutoZOOM | 51.04% | 30032 | 52.36% | 28547 | 60.00% | 25836 | 53.78% | 28356 |
| P-RGF | 99.17% | 3704 | 98.05% | 5498 | 97.96% | 5769 | 98.17% | 6896 |
| Trans-P-RGF | 99.58% | 662 | 99.31% | 1896 | 99.05% | 2267 | 99.16% | 3192 |
| TREMBA | **99.72%** | **285** | **99.44%** | **443** | **99.72%** | **224** | **99.72%** | **422** |

settings are the same in Appendix A.2. As shown in Table 9, this autoencoder is even worse than the original autoencoder despite small dimension of the embedding space. In addition, we also changed to dimension of the data-dependent prior of Trans-P-RGF to match the dimension of TREMBA, whose performance is also not better than before. They show that simply diminishing the size of the embedding space may not lead to better performance. The performance gain of TREMBA comes beyond the effect of diminishing the dimension of the embedding space.

## A.8 EXAMPLES OF ATTACKING GOOGLE CLOUD VISION API

Figure 11 shows one example of attacking Google Cloud Vision API. TREMBA successfully make the shark to be classified as green. Compared with Trans-NES$_{PGD}$, TREMBA hugely changes the

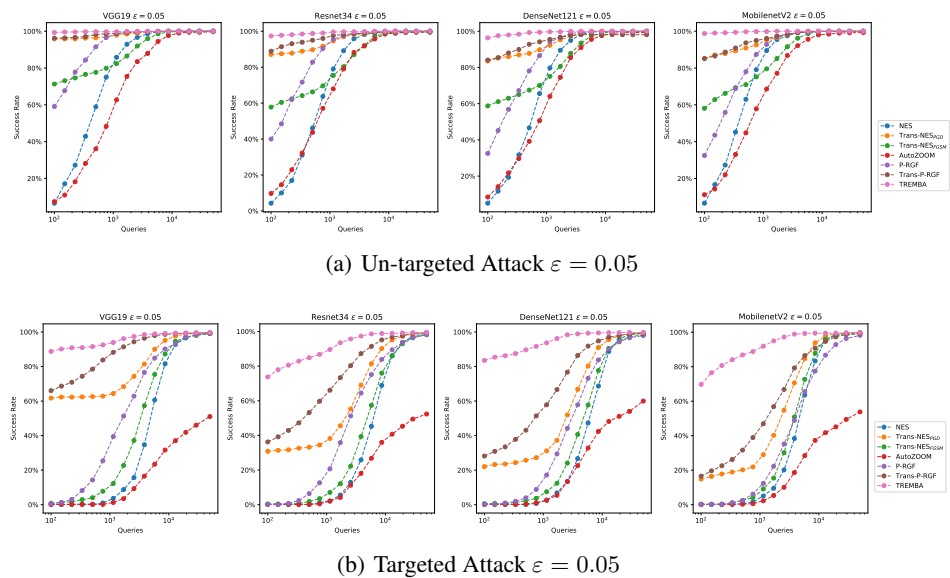

(a) Un-targeted Attack $\varepsilon = 0.05$

(b) Targeted Attack $\varepsilon = 0.05$

Figure 8: We show the success rate at different query levels for targeted and un-targeted attack at $\varepsilon = 0.05$ for ImageNet.

Table 8: Hyperparameter sweep over $b$ on Densenet121 for un-targeted attack on ImageNet

| Sweep over $b$ | $b = 10$ | | $b = 30$ | | $b = 40$ | | $b = 50$ | |
|---|---|---|---|---|---|---|---|---|
| | Success | Queries | Success | Queries | Success | Queries | Success | Queries |
| NES | 100% | 1323 | 100% | 1284 | 100% | 1433 | 100% | 1639 |
| Trans-NES$_{PGD}$ | 100% | 915 | 100% | 791 | 100% | 707 | 100% | 639 |
| Trans-NES$_{FGSM}$ | 100% | 1037 | 100% | 916 | 100% | 879 | 100% | 886 |
| AutoZOOM | 90.9% | 6052 | 96.2% | 4148 | 97.1% | 4066 | 97.3% | 4366 |
| P-RGF | 99.73% | 717 | 99.86% | 860 | 99.86% | 949 | 99.86% | 1095 |
| Trans-P-RGF | 98.50% | 1139 | 99.86% | 479 | 99.86% | 487 | 100% | 427 |
| TREMBA | 100% | 150 | 100% | 205 | 100% | 274 | 100% | 299 |

Table 9: Change of dimension of the embedding space of AutoZOOM. The task is un-targeted attack on ImageNet.

| Attack | VGG19 | | Resnet34 | | DenseNet121 | | MobilenetV2 | |
|---|---|---|---|---|---|---|---|---|
| | Success | Queries | Success | Queries | Success | Queries | Success | Queries |
| AutoZOOM | 64.35% | 19684 | 71.94% | 16931 | 68.44% | 17871 | 71.15% | 16134 |
| Trans-P-RGF | 99.86% | 194 | 99.58% | 508 | 99.59% | 610 | 99.58% | 705 |

labels of the image. It is hard to say the overall classification of Trans-NES$_{PGD}$ is wrong. However, the labels of TREMBA are definitely not correct.

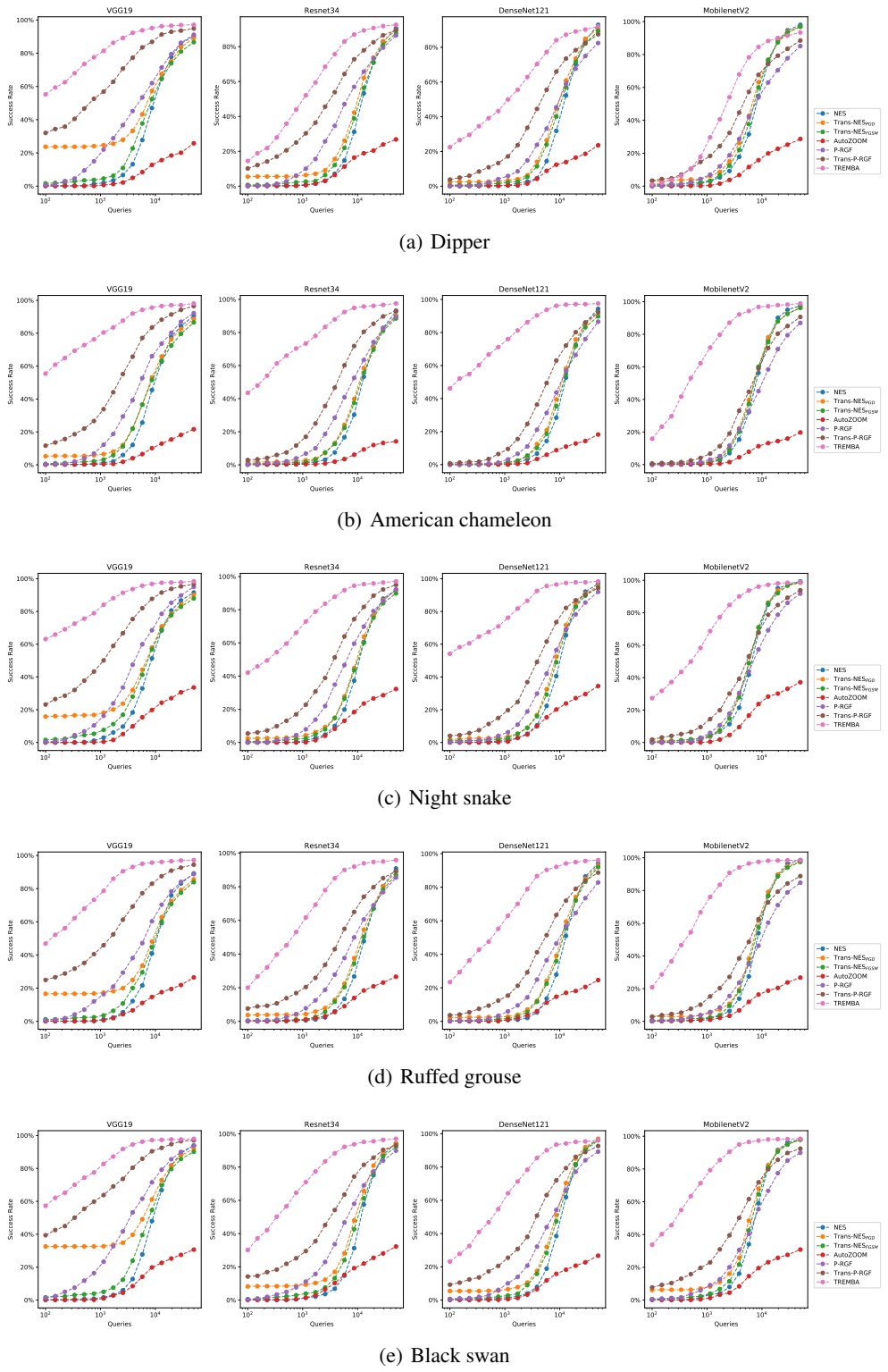

Figure 9: The success rate at different query levels for attack targeted at different class. Targeted classes are: (a)Dipper; (b)American chameleon; (c)Night snake; (d)Ruffed grouse; (e)Black swan

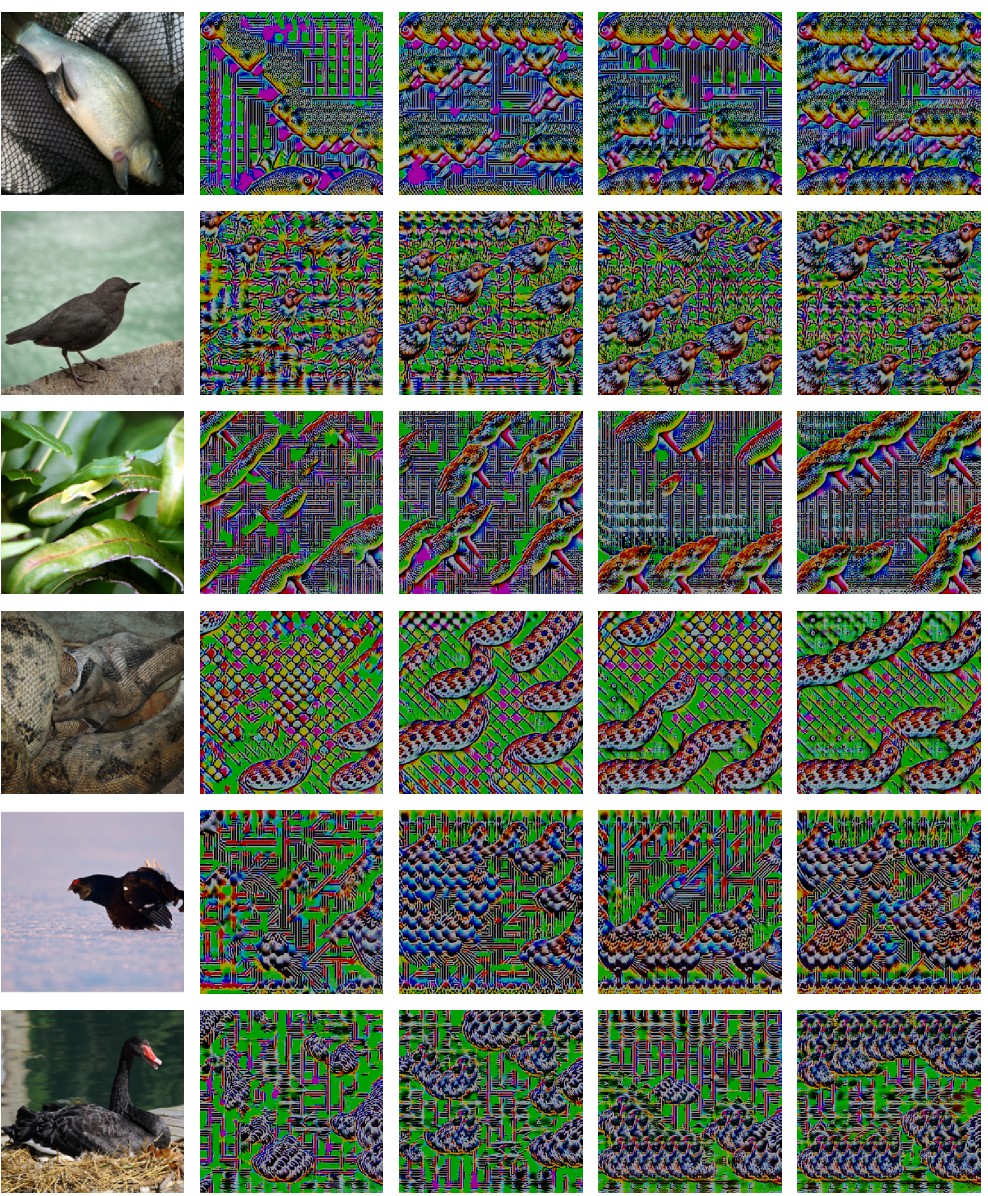

Figure 10: Visualization of adversarial perturbations for targeted attack on ImageNet. The first column shows one example of the target class. Other columns show the adversarial perturbations.

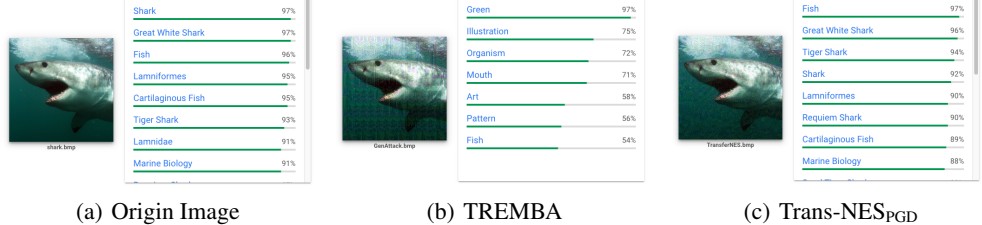

(a) Origin Image      (b) TREMBA      (c) Trans-NES$_{PGD}$

Figure 11: One example of adversarial image for attacking Google Cloud Vision API

Table 10: Comparision between CombOpt and TREMBA for un-targeted attack on Imagenet.

| Attack | VGG19 | | Resnet34 | | DenseNet121 | | MobilenetV2 | |
|---|---|---|---|---|---|---|---|---|
| | Success | Queries | Success | Queries | Success | Queries | Success | Queries |
| CombOpt | **100%** | 567 | **100%** | 499 | **100%** | 569 | **100%** | 522 |
| TREMBA | **100%** | **88** | **100%** | **183** | **100%** | **172** | **100%** | **61** |

Table 11: Comparision between CombOpt and TREMBA for targeted attack on Imagenet.

| Attack | VGG19 | | Resnet34 | | DenseNet121 | | MobilenetV2 | |
|---|---|---|---|---|---|---|---|---|
| | Success | Queries | Success | Queries | Success | Queries | Success | Queries |
| CombOpt | 93.76% | 9767 | 94.86% | 8024 | 97.41% | 6970 | 96.92% | 8575 |
| TREMBA | **98.47%** | **853** | **96.38%** | **1206** | **98.50%** | **1124** | **99.16%** | **1210** |

## A.9 COMPARISION BETWEEN TREMBA AND COMBOPT

CombOpt is one of the SOTA score-based black-box attack. We compared our method with it on the targeted and un-targeted attack on Imagenet. The targeted attack is 0 and $\varepsilon = 0.03125$. As shown in Table 10 and Table 11, TREMBA requires much lower queries than CombOpt. It demonstrates the great improvement by combining the transfer-based and score-based attack.

## B ARCHITECTURE OF CLASSIFIERS AND GENERATORS

### B.1 CLASSIFIER

Table 12: Model architectures for the MNIST

| ConvNet1 | ConvNet2 | FCNet |
|---|---|---|
| Conv(64, 5, 5)+ReLU | Conv(16, 3, 3)+ReLU | FC(512)+ReLU |
| MaxPool(2,2) | Conv(16, 3, 3)+ReLU | FC(10)+Softmax |
| Conv(64, 5, 5)+ReLU | MaxPool(2,2) | |
| MaxPool(2,2) | Conv(32, 3, 3)+ReLU | |
| Dropout(0.25) | Conv(32, 3, 3)+ReLU | |
| FC(128)+ReLU | Conv(32, 3, 3)+ReLU | |
| Dropout(0.5) | MaxPool(2,2) | |
| FC(10)+Softmax | FC(512)+ReLU | |
| | FC(10)+Softmax | |

Table 12 lists the architectures of ConvNet1, ConvNet2 and FCNet. The architecture of ResNeXt used in CIFAR10 is from `https://github.com/prlz77/ResNeXt.pytorch`. We set the depth to be 20, the cardinality to be 8 and the widen factor to be 4. Other architectures of classifiers are specified in the corresponding paper.

### B.2 GENERATOR

Table 13 lists the architectures of generator for three datasets. For AutoZOOM, we find our architectures are not suitable and use the same generators in the corresponding paper.

## C HYPERPARAMETERS

### C.1 TRAINING GENERATOR

We trained the generators with learning rate starting at 0.01 and decaying half every 50 epochs. The whole training process was 500 epochs. The batch size was determined by the memory of GPU. Specifically, we set batch size to be 256 for MNIST and CIFAR10 defense model, 64 for ImageNet model. All large $\kappa$ will work well for our method and we chose $\kappa = 200.0$. All the experiments were performed using *pytorch* on NVIDIA RTX 2080Ti.

Table 13: Architectures of encoder and decoder. ConvReLUBN and DeconvReLUBN represent convolution or deconvolution followed by ReLU and batch normalization. The parameters $(c, m, n)$ used in ConvReLUBN or DeconvReLUBN mean $c$ channels with $m \times n$ kernel size. MaxPool$(m, n)$ represents max pooling with $(m, n)$ kernel size and $(m, n)$ stride.

|  | MNIST | CIFAR10 | ImageNet |
|---|---|---|---|
| Encoder | ConvReLUBN(16,3,3) | ConvReLUBN(16,3,3) | ConvReLUBN(16,3,3) |
|  | ConvReLUBN(32,3,3) | ConvReLUBN(32,3,3) | ConvReLUBN(32,3,3) |
|  | ConvReLUBN(32,3,3) | ConvReLUBN(32,3,3) | MaxPool(2,2) |
|  | MaxPool(2,2) | MaxPool(2,2) | ConvReLUBN(64,3,3) |
|  | ConvReLUBN(32,3,3) | ConvReLUBN(32,3,3) | ConvReLUBN(64,3,3) |
|  | ConvReLUBN(16,3,3) | ConvReLUBN(32,3,3) | MaxPool(2,2) |
|  | ConvReLUBN(2,3,3) | ConvReLUBN(8,3,3) | ConvReLUBN(128,3,3) |
|  | MaxPool(2,2) | MaxPool(2,2) | ConvReLUBN(128,3,3) |
|  |  |  | MaxPool(2,2) |
|  |  |  | ConvReLUBN(32,3,3) |
|  |  |  | ConvReLUBN(8,3,3) |
|  |  |  | MaxPool(2,2) |
| Decoder | ConvReLUBN(32,3,3) | ConvReLUBN(32,3,3) | ConvReLUBN(32,3,3) |
|  | ConvReLUBN(32,3,3) | ConvReLUBN(32,3,3) | DeconvReLUBN(64,3,3) |
|  | DeconvReLUBN(64,3,3) | DeconvReLUBN(64,3,3) | ConvReLUBN(128,3,3) |
|  | ConvReLUBN(64,3,3) | ConvReLUBN(64,3,3) | DeconvReLUBN(128,3,3) |
|  | ConvReLUBN(64,3,3) | ConvReLUBN(64,3,3) | ConvReLUBN(128,3,3) |
|  | DeconvReLUBN(16,3,3) | DeconvReLUBN(16,3,3) | DeconvReLUBN(64,3,3) |
|  | Conv(1,1,1) | Conv(3,1,1) | ConvReLUBN(32,3,3) |
|  |  |  | DeconvReLUBN(16,3,3) |
|  |  |  | ConvReLUBN(3,1,1) |

## C.2 EVALUATION

Table 14 to 19 list the hyperparameters for all the algorithms. The learning rate was fine-tuned for all the algorithms. We set sample size $b = 20$ for all the algorithms for fair comparisons.

Table 14: Hyperparameters for NES

|  | MNIST | CIFAR10 | ImageNet | | |
|---|---|---|---|---|---|
|  |  |  | Un-targeted | Targeted | Un-targeted Defense |
| Sample size ($b$) | 20 | 20 | 20 | 20 | 20 |
| Learning rate ($\eta$) | 0.2 | 0.05 | 0.1 | 0.05 | 0.1 |

Table 15: Hyperparameters for Trans-NES$_{\text{PGD}}$ and Trans-NES$_{\text{FGSM}}$. White-box iteration, white-box margin and white-box learning rate mean the hyperparameters for generating the starting point on the source network for Trans-NES$_{\text{PGD}}$.

|  | MNIST | CIFAR10 | ImageNet | | |
|---|---|---|---|---|---|
|  |  |  | Un-targeted | Targeted | Un-targeted Defense |
| Sample size ($b$) | 20 | 20 | 20 | 20 | 20 |
| Learning rate ($\eta$) | 0.2 | 0.05 | 0.1 | 0.05 | 0.1 |
| White-box iteration | 50 | 100 | 50 | 50 | 100 |
| White-box margin($\kappa$) | 100 | 100 | 100 | 100 | 100 |
| White-box learning rate | 0.05 | 0.1 | 0.01 | 0.005 | 0.1 |

Table 16: Hyperparameters for AutoZOOM.

| | MNIST | CIFAR10 | ImageNet | | |
| --- | --- | --- | --- | --- | --- |
| | | | Un-targeted | Targeted | Un-targeted Defense |
| Sample size ($b$) | 20 | 20 | 20 | 20 | 20 |
| Learning rate ($\eta$) | 5.0 | 20.0 | 5.0 | 3.0 | 5.0 |

Table 17: Hyperparameters for P-RGF and Trans-P-RGF.

| | MNIST | CIFAR10 | ImageNet | | |
| --- | --- | --- | --- | --- | --- |
| | | | Un-targeted | Targeted | Un-targeted Defense |
| Sample size ($b$) | 20 | 20 | 20 | 20 | 20 |
| Learning rate ($\eta$) | 0.1 | 0.05 | 0.005 | 0.003 | 0.005 |
| White-box iteration | 50 | 100 | 50 | 50 | 100 |
| White-box margin($\kappa$) | 100 | 100 | 100 | 100 | 100 |
| White-box learning rate | 0.05 | 0.1 | 0.01 | 0.01 | 0.1 |

Table 18: Hyperparameters for TREMBA.

| | MNIST | CIFAR10 | ImageNet | | |
| --- | --- | --- | --- | --- | --- |
| | | | Un-targeted | Targeted | Un-targeted Defense |
| Sample size ($b$) | 20 | 20 | 20 | 20 | 20 |
| Learning rate ($\eta$) | 0.3 | 2.0 | 5.0 | 3.0 | 5.0 |

Table 19: Hyperparameters for TREMBA$_{OSP}$ .

| | CIFAR10 Defense | ImageNet Defense |
| --- | --- | --- |
| Sample size ($b$) | 20 | 20 |
| Learning rate ($\eta$) | 2.0 | 5.0 |
| White-box iteration | 100 | 100 |
| White-box margin($\kappa$) | 100 | 100 |
| White-box learning rate | 1.0 | 2.0 |

