# OpenReview forum: "Black-Box Adversarial Attack with Transferable Model-based Embedding"
_ICLR.cc/2020/Conference — Accept (Poster)_

### Official Review · AnonReviewer3 · 2019-10-23
**Official Blind Review #3**

**Rating:** 6

**Review:**

This paper proposed a new method for black-box adversarial attacks which tries to learn a  low-dimensional embedding using a pretrained model and then performs efficient search within the embedding space to attack the target network. The proposed method can produce perturbation with semantic patterns are easily transferable. It can be used to improve the query efficiency in black-box attacks.

- The main idea of this paper is quite simple, i.e., using a autoencoder model to capture encoding in the embedding space and then searching over the embedding space for possible attacks. While searching for adversarial examples in the embedding space is not something new, such as manifold attack or GAN-based attack, the authors claimed that by doing so, it can help reduce the query complexity of black-box attacks. However, it is not immediately clear to me why this can help reduce the query complexity, as intuitively, restricting the attack image space to an embedding space (or a manifold) will naturally increase the difficulty for finding adversarial examples. The authors’ explanation is not quite convincing to me since many adversarial examples are not necessarily on the embedding space.

- Algorithm 1 is not clearly written and I do not understand the update rule in Algorithm 1. What is Li exactly? If Li means eq(1) or eq(2), it seems totally independent of sampled Guassian noise? Also, the update rule in Line 5 of Algorithm 1 is different from what is described in eq(4) or other black-box attack algorithms. Can the author explain the algorithm design with details?

- In experiments section, the authors miss a few important black-box attack baselines. I would suggest the authors to further comment and compare with the following black-box attacks to better demonstrate the performance of the proposed algorithm.

Ilyas, Andrew, Logan Engstrom, and Aleksander Madry. "Prior convictions: Black-box adversarial attacks with bandits and priors." ICLR 2019.
Moon, Seungyong, Gaon An, and Hyun Oh Song. "Parsimonious Black-Box Adversarial Attacks via Efficient Combinatorial Optimization." ICML 2019.
Chen, Jinghui, Jinfeng Yi, and Quanquan Gu. "A Frank-Wolfe Framework for Efficient and Effective Adversarial Attacks." arXiv preprint arXiv:1811.10828 (2018).

- Also can the authors further conduct experiments using more common choice of \epsilon to help the reader get better understandings. For example, add ImageNet experiments with \epsilon = 0.05.

Detailed comments:

- In section 3.2, the author suggests that by removing the sign function, the attacks can be more effective in (Li et al., 2019).  I didn’t find the corresponding argument in (Li et al., 2019). Can the authors be more specific on this argument?

-------------------------
I have read the response and it addressed my concerns. I will increase my score

**Experience Assessment:**

I have published in this field for several years.

**Review Assessment: Checking Correctness Of Derivations And Theory:**

I carefully checked the derivations and theory.

**Review Assessment: Checking Correctness Of Experiments:**

I carefully checked the experiments.

**Review Assessment: Thoroughness In Paper Reading:**

I read the paper thoroughly.

---

> ### Author Response · Authors · 2019-11-11
> **Response**
>
> Thank you for the comments. We address your main concerns as follows.
>
> 1. To answer your argument that "intuitively, restricting the attack image space to an embedding space (or a manifold) will naturally increase the difficulty for finding adversarial examples". The main reason is about the efficient in finding an adversarial example when the number of queries is limited when doing the adversarial attack comparisons. For exmaple, if we only consider no more than 50000 queries (as in our experiments), then it is much easier to find a perturbation in the embedding space simply because a randomly chosen point in the embedding space is much more likely to be a successful adversarial attack example. Although the orginal space may contain additional adversarial examples, a randomly selected point that is outside the embedding space is much less likely to be an adversarial attack example, so it is significantly more difficult to use 50000 queries to find an adversarial attack example in the larger space, which lowered the success rate reported in the paper for such methods. In short, it is simply more efficient to look for adversarial examples in the restricted space.
>
> In fact, previous methods such as BanditTD [1] and Parsimonious Attack [2] have also restrcted the search space of adversarial examples to achieve better efficiency. BanditTD restricted the search space by incorporating the data prior. Parsimonious Attack restricted the search space to $\{-\epsilon,\epsilon\}$. In both cases, the adversarial examples are likely to be found in the restricted search space, and smaller dimension leads to faster zeroth order optimization. In TREMBA, we train the generator to produce adversarial examples for the source model. As we shown in the Appendix A.3, the generator find some structural patterns that can easily fool the source model. With high transferability of the adversarial attack, these patterns are easy to transfer to a new model. The adversarial examples have higher concentration in the embedding space than in the original image space. With the higher density of the adversarial perturbtions in the embedding space and the smaller dimension of the embedding space, TREMBA can find adversarial examples more efficiently.
>
> 2. Sorry for a typo in the Algorithm 1. In line 5 of the algorithm, we should add $\nabla_{z_{t-1}} \log \mathcal{N}(\nu_i|z_{t-1}, \sigma^2)$ behind $L_i$, and it becomes $z_t = z_{t-1} - \frac{\eta}{b}\sum L_i\nabla_{z_{t-1}} \log \mathcal{N}(\nu_i|z_{t-1}, \sigma^2)$. This way, line 5 becomes consistent with eq (4). $L_i$ refers to the loss functions in eq (1) and eq (2) for un-targeted attack and targeted attack respectively.
>
> 3. Thanks for mentioning the additional interesting references we missed. We have added references and comparisons in the revised paper. Please be aware that these methods did not use the information of the source model, therefore comparisons between their methods and TREMBA would not be completely fair. Results in these papers with the larger $\epsilon=0.05$ require average queries much more than those of TREMBA with smaller $\epsilon$. Based on your recommendation, in the updated version of the paper, we have included "Parsimonious Attack" for comparison in the Appendix A.9 because it seems to achieve the lowest number of queries among these three methods. We would also like to point out that we compare TREMBA with P-RGF [4], which also incorporates the data prior with performance surpassing BanditTD.
>
> 4. We have added results for $\epsilon=0.05$ in the updated paper (see Appendix A.5). We did not use this $\epsilon$ in our paper because such a large $\epsilon$ leads to very high success rates for transfer-based methods.
>
> 5. The argument is shown in the previous version of the paper: https://openreview.net/forum?id=ryeoxnRqKQ. In the experiment, they tried to remove the PGD step and the success rate of attacking THERM-ADV and SAP was increased. We have updated the citation in the Section 3.2.
>
> [1] Ilyas, Andrew, Logan Engstrom, and Aleksander Madry. "Prior convictions: Black-box adversarial attacks with bandits and priors." ICLR 2019.
> [2] Moon, Seungyong, Gaon An, and Hyun Oh Song. "Parsimonious Black-Box Adversarial Attacks via Efficient Combinatorial Optimization." ICML 2019.
> [3] Chen, Jinghui, Jinfeng Yi, and Quanquan Gu. "A Frank-Wolfe Framework for Efficient and Effective Adversarial Attacks." arXiv preprint arXiv:1811.10828 (2018).
> [4] Cheng, Shuyu, et al. "Improving Black-box Adversarial Attacks with a Transfer-based Prior." arXiv preprint arXiv:1906.06919 (2019).

---

### Official Review · AnonReviewer2 · 2019-10-23
**Official Blind Review #2**

**Rating:** 8

**Review:**

Review: The paper proposes a new framework (TREMBA) for black-box adversarial attack. The method utilizes a pretrained source network to learn a low dimensional embedding, it then searches efficiently within the embedding space (using NES) and produces an adversarial perturbation that can attack an unknown target network. A generator model first encodes an input to a latent vector and then decodes it to give an adversarial perturbation as an output. This generator is trained so that it can fool the source network and is then used to find the adversarial pattern when searching in the latent space. TREMBA produces perturbations with high level semantic patterns, and is easily transferable to different target architectures. The paper demonstrates its performance in terms of number of queries vs success rate on different datasets, Google cloud vision API and adversarially defended networks.

- I like the exhaustive evaluation and comparative study done in the paper. It was especially interesting to see how TREMBA outperforms other techniques when attacking SOTA defended networks (on CIFAR 10 and Imagenet dataset).
- When the method seems intuitive, it shows a novel way to combine transfer-based and score-based attack methods.
- The motivation behind using low dim embedding space to accelerate adversarial pattern searching is also well explained in the paper.
- The contributions are well-stated in the paper and definitely show an improvement over the past methods in not only reducing the number of queries but also improving the success rate of the attack.

Comments:
The loss function L_{target}(xi, t) on Page 3 has yi instead of t in the equation.


**Experience Assessment:**

I have read many papers in this area.

**Review Assessment: Checking Correctness Of Derivations And Theory:**

I assessed the sensibility of the derivations and theory.

**Review Assessment: Checking Correctness Of Experiments:**

I carefully checked the experiments.

**Review Assessment: Thoroughness In Paper Reading:**

I read the paper thoroughly.

---

> ### Author Response · Authors · 2019-11-11
> **Response**
>
> Thank you for your comments. We appreciate them,  and have fixed the typo you pointed out in the revised paper.

---

### Official Review · AnonReviewer1 · 2019-10-30
**Official Blind Review #1**

**Rating:** 6

**Review:**

This paper proposes a new black-box adversarial attack method called TREMBA, in which the search for the “adversary” is done in a reduced space z. Summary of its contributions:
-	A attack method that improves query efficiency of black-box attack
-	Produces perturbations that are effective across different networks
-	Improves attack success over SOTA defended networks

In general, the paper is very well written, with clear mostly clear exposition and sufficient experimental verification. What follows are the itemized pros and cons (mostly just points that would be good to address):

[pros]:
-	Well written
-	A good overview of previous methods and how the TREMBA fits within them
-	Sufficient experimental validation

[points to address]
-	In Black-Box Attack method of related works, did you mean to say, “Targeted attack is much harder than un-targeted attack for transfer-based method.”?
-	You ought to explain what NES is - Natural Evolution Strategies – and the general description of the method, as it is a major part of your algorithm (Section 3.2). It took two papers to find what NES stands for.
-	In section 3.2 you write – “The sign function provides an approximation of the gradient, …” – is there a citation that should go with this claim?
-	Make sure to explain all of the variables in the paper, e.g. $\omega_k$ in Eq. (3) or $\nu_k$ in Eq. (4).
-	IS there a particular reason you chose the hinge loss to train the generator? Could you have used other losses instead?
-	“A higher value of $\kappa$ laeds to higher transferability to other models” – maybe a citation required? Or else more intuition?
-	Adding specifics about ConvNet1, ConvNet2
-	How did you set $\epsilon$ ?
-	Would changing the sample size for each method improve the performance for respective methods?
-	It would have been interesting to include a Future Works section
-	A more thorough discussion why the models works so well.


**Experience Assessment:**

I do not know much about this area.

**Review Assessment: Checking Correctness Of Derivations And Theory:**

I assessed the sensibility of the derivations and theory.

**Review Assessment: Checking Correctness Of Experiments:**

I assessed the sensibility of the experiments.

**Review Assessment: Thoroughness In Paper Reading:**

I read the paper thoroughly.

---

> ### Author Response · Authors · 2019-11-11
> **Response**
>
> Thanks for the comments. Please find our responses below.
>
> 1. For transfer-based method, targeted attack is believed to be harder than the untargeted attack in [1]. They also found that using an ensemble of networks improves the transferability of adversarial perturbtions for targeted attack. We have added the citation in the related work about black-box attack.
>
> 2. We have added more detailed descriptions of NES in our updated paper on page 4.
>
> 3. The sign function has been widely used in adversarial attack in both white-box attack [2] and black-box attack [3]. We have added citations on page 4.
>
> 4. We have added more explanations in our paper (See Section 3.2)
>
> 5. The use of hinge loss was proposed in the C&W attack [4], and it was also widely used in black-box attack [5,6]. We just follow the convention and use the hinge loss in our training and attack.  Other loss functions such as the cross entropy loss can also be applied, and we believe the corresponding results will be similar to that of the hinge loss. We added a remark in Section 3.1.
>
> 6. This was pointed out in the C&W attack [4]. They showed in their experiment that as $\kappa$ increases, the success rate of transfer attack also increases. We added a citation about this on page 3.
>
> 7. We showed architectures of ConvNet1 and ConvNet2 in Appendix B.
>
> 8. We chose $\epsilon=0.03125$ because it is commonly used. We performed experiments other $\epsilon$ values such as 0.02 and 0.04. For another commonly used value $\epsilon=0.05$, TREMBA can also attack most images successfully. As requested by reviewer 3, we have added experiements of $\epsilon=0.05$ in the updated version (see Appendix A.5)
>
> 9. As we showed in Appendix A.6, the performance will not change too much as we change the sample size. As the sample size decreases, the gradient estimation will be more noisy and it will take more steps to find the adversarial example.
>
> 10. We have added a comment on future works in our conclusion section.
>
> 11. As we discussed in section 3.2, we believe TREMBA works so well because of the high density of the adversarial perturbtion in the embedding space and the small dimension of the embedding space. The high level semantic adversarial patterns found by the generator can be easily transferred to other networks. Therefore the adversarial examples have significant more concentration in the embedding space than in the original image space. In addition to the denser concentration of the adversarial perturbations in the embedding space, its smaller dimension also makes zeroth order optimization faster. If you feel additional information is needed, we'd be happy to elaborate more.
>
>
> [1] Liu, Yanpei, et al. "Delving into transferable adversarial examples and black-box attacks." arXiv preprint arXiv:1611.02770.
> [2] Madry, Aleksander, et al. "Towards deep learning models resistant to adversarial attacks." arXiv preprint arXiv:1706.06083 (2017).
> [3] Ilyas, Andrew, et al. "Black-box adversarial attacks with limited queries and information." arXiv preprint arXiv:1804.08598 (2018).
> [4] Carlini, Nicholas, and David Wagner. "Towards evaluating the robustness of neural networks." 2017 IEEE Symposium on Security and Privacy (SP). IEEE, 2017.
> [5] Chen, Pin-Yu, et al. "Zoo: Zeroth order optimization based black-box attacks to deep neural networks without training substitute models." Proceedings of the 10th ACM Workshop on Artificial Intelligence and Security. ACM, 2017.
> [6] Li, Yandong, et al. "NATTACK: Learning the Distributions of Adversarial Examples for an Improved Black-Box Attack on Deep Neural Networks." arXiv preprint arXiv:1905.00441 (2019).

---

### Decision · Program_Chairs · 2019-12-19

**Decision:**

Accept (Poster)

**Comment:**

This paper proposes a new black-box adversarial attack approach which learns a low-dimensional embedding using a pretrained model and then performs efficient search in the embedding space to attack target networks. The proposed approach can produce perturbation with semantic patterns that are easily transferable and improve the query efficiency in black-box attacks. All reviewers are in support of the paper after author response. I am very happy to recommend accept.